# Cinnamides Target *Leishmania amazonensis* Arginase Selectively

**DOI:** 10.3390/molecules25225271

**Published:** 2020-11-12

**Authors:** Edson Roberto da Silva, Júlio Abel Alfredo dos Santos Simone Come, Simone Brogi, Vincenzo Calderone, Giulia Chemi, Giuseppe Campiani, Trícia Maria Ferrreira de Sousa Oliveira, Thanh-Nhat Pham, Marc Pudlo, Corine Girard, Claudia do Carmo Maquiaveli

**Affiliations:** 1Laboratório de Farmacologia e Bioquímica (LFBq), Departamento de Medicina Veterinária, Universidade de São Paulo Faculdade de Zootecnia e Engenharia de Alimentos, Pirassununga, SP 13635-900, Brazil; jcome84@gmail.com (J.A.A.d.S.S.C.); tricia@usp.br (T.M.F.d.S.O.); 2Departamento de Pré-Clínicas, Universidade Eduardo Mondlane, Faculdade de Veterinária, Av. de Moçambique, Km 1.5, Maputo CP 257, Mozambique; 3Department of Pharmacy, University of Pisa, via Bonanno 6, 56126 Pisa, Italy; vincenzo.calderone@unipi.it; 4Department of Biotechnology, Chemistry, and Pharmacy, DoE Department of Excellence 2018–2022 Università degli Studi di Siena via Aldo Moro 2, 53100 Siena, Italy; GChemi001@dundee.ac.uk (G.C.); giuseppe.campiani@unisi.it (G.C.); 5PEPITE EA4267, University Bourgogne Franche-Comté, F-25000 Besançon, France; thanhata1@yahoo.com (T.-N.P.); marc.pudlo@univ-fcomte.fr (M.P.)

**Keywords:** arginase, polyamines, *Leishmania*

## Abstract

Caffeic acid and related natural compounds were previously described as *Leishmania amazonensis* arginase (L-ARG) inhibitors, and against the whole parasite in vitro. In this study, we tested cinnamides that were previously synthesized to target human arginase. The compound caffeic acid phenethyl amide (CAPA), a weak inhibitor of human arginase (IC_50_ = 60.3 ± 7.8 μM) was found to have 9-fold more potency against L-ARG (IC_50_ = 6.9 ± 0.7 μM). The other compounds that did not inhibit human arginase were characterized as L-ARG, showing an IC_50_ between 1.3–17.8 μM, and where the most active was compound **15** (IC_50_ = 1.3 ± 0.1 μM). All compounds were also tested against *L. amazonensis* promastigotes, and only the compound CAPA showed an inhibitory activity (IC_50_ = 80 μM). In addition, in an attempt to gain an insight into the mechanism of competitive L-ARG inhibitors, and their selectivity over mammalian enzymes, we performed an extensive computational investigation, to provide the basis for the selective inhibition of L-ARG for this series of compounds. In conclusion, our results indicated that the compounds based on cinnamoyl or 3,4-hydroxy cinnamoyl moiety could be a promising starting point for the design of potential antileishmanial drugs based on selective L-ARG inhibitors.

## 1. Introduction

Leishmaniasis is a neglected tropical disease caused by protozoan parasites from more than 20 *Leishmania* species. The main types of disease are visceral leishmaniasis (VL) (known as kala-azar), cutaneous leishmaniasis (CL), and mucocutaneous leishmaniasis (MCL) [1]. More than 90% of new cases of disease reported in the last four years occurred in Brazil, Ethiopia, Somalia, Sudan, and India. Although CL is the most common form of the disease, with about 5000 new cases reported in the endemic regions in 2015, VL causes more than 20,000 deaths/year [1].

The search for more effective treatments for leishmaniasis remains a challenge [2]. Pentavalent antimonial compounds [3], amphotericin B [4], pentamidine [5], and miltefosine [6] have been used in the treatment of leishmaniasis, with different efficacy. High toxicity associated with drug resistance [7], and leishmaniasis–HIV co-infection [8] are the main problems causing the treatment failure and poor efficacy of the drugs in use.

In search of new targets, the arginase enzyme was established as a potential drug target candidate to develop new antileishmanial drugs. Arginase (EC 3.5.3.1) is a trimeric manganese-containing enzyme that hydrolyzes the amino acid L-arginine into L-ornithine and urea. L-ornithine is used to synthesize polyamines, and is essential to parasite growth and host infection [9,10]. Polyamines are also essential to trypanothione production, an important antioxidant agent in the control of reactive oxygen species (ROS) [11]. In fact, arginase expression and activity in *Leishmania* contribute to higher parasite infectivity, and play a major role in the pathogenicity of the infection [12]. An increase of arginase activity decreases the availability of L-arginine to nitric oxide synthase (NOS), and reduces NO formation and uncouples NOS, reducing the host’s defensive capacity, and increasing the parasite infectivity [13].

Caffeic acid and derived compounds, such as chlorogenic and rosmarinic acids, were previously described against *L. amazonensis* promastigotes and amastigotes [14]. These compounds were also described as good *L. amazonensis* arginase (L-ARG) inhibitors [15]. Previously, a series of nineteen cinnamide derivatives were designed, synthesized, and evaluated for their inhibitory activity against mammalian arginase. The study showed that bovine arginase (B-ARG) inhibition was higher than 50% for eleven compounds at 100 µM, and resulted in the selection of the caffeic acid phenethylamide (CAPA) compound, which obtained good results in the in vitro inhibition of B-ARG, but did not inhibit human arginase [16]. In this study, we tested the cinnamides designed as mammalian arginase inhibitors, containing a catechol group potentially responsible for a selective L-ARG inhibition, as observed by molecular docking, also highlighting possible interactions of competitive arginase inhibitors. Furthermore, the compounds were screened against the whole parasite in vitro.

## 2. Results

### 2.1. Arginase Inhibition and Antileishmanial Activity

A set of 10 cinnamide derivatives (Table 1) were tested for the inhibition of *Leishmania amazonensis* recombinant arginase (L-ARG). The half concentration inhibitory (IC_50_), maximum effect (E_max_), constant of enzyme dissociation (K_i_ and K_is_), and the mechanism of enzyme inhibition were determined. Analysis of the drug concentration–response plot was used to determine the IC_50_. The IC_50_ varies from 1.3 ± 0.1 µM (compound **15**) to 17.8 ± 3.2 µM (compound **17**). Compounds **11** and **13–15** show the best IC_50_ values (Table 1).

The kinetics of enzyme inhibition reveals that cinnamide derivatives present different mechanisms of inhibition: compounds CAPA, **6**, **13**, **14**, and **17** are competitive inhibitors (K_i_ <<< K_is_), compounds **15**, **18,** and **19** are mixed (K_i_ ≠ K_is_) inhibitors, compound **11** is uncompetitive (K_i_ >>> K_is_), and compound **12** is a noncompetitive inhibitor (K_i_ = K_is_). Analyzing Ki values, compounds **14** and **15** showed better affinities to the enzyme. However, the results did not differ significantly (*p* > 0.005) from compounds CAPA, **13**, and **18**. Compounds **12**, **17**, and **19**, showed the lowest affinity to the enzyme. Moreover, we evaluated compounds for their inhibitory activity against B-ARG. Results are reported in Table 2.

Despite the high affinity of the compounds **14** and **15** to parasite arginase, only the compound CAPA was able to kill the promastigote parasite with an IC_50_ of 82.56 µM, and a 95% confidence interval from 80.15 to 85.00 µM (r^2^ = 0.98).

### 2.2. Computational Studies

To gain insight into the binding modes of the competitive inhibitors (CAPA, **6**, **13**, **14**, and **17**) of L-ARG we performed an in silico investigation based on molecular docking coupled to the evaluation of ligand binding energy, as previously described [15,17,18]. Moreover, the significant differences, regarding the above mentioned inhibitors, in inhibiting L-ARG with respect to the mammalian arginase (bovine), prompted us to design a series of additional studies, in order to understand the different behavior of the compounds. Accordingly, we built a homology model of B-ARG that was employed, first in a detailed analysis of the binding site in comparison with L-ARG, and subsequently in further molecular docking studies combined with the evaluation of ligand binding energy. Furthermore, one of the most promising competitive L-ARG inhibitors (compound **14**), showing the highest selectivity index compared to the IC_50_ against B-ARG, was submitted to an extensive molecular dynamics (MD) investigation for a better understanding of its selectivity against L-ARG.

#### Molecular Docking Studies, Homology Modeling, Binding Sites Analysis, and Molecular Dynamics Simulation

The binding modes of the competitive inhibitors of L-ARG (CAPA, **6**, **13**, **14** and **17**) were investigated, employing an in silico protocol, and combining a molecular docking simulation with the evaluation of ligand binding energy. Notably, we preferred to focus the computational studies only on the binding of the competitive inhibitors of L-ARG, given that for the other inhibitors, showing a uncompetitive or mixed mode of inhibition, the mechanism is not completely understood, and a discussion about the binding of these compounds to L-ARG could be too speculative. For carrying out the computational experiments we used Glide (Glide, Schrödinger, LLC, New York, NY, USA, 2016) and Prime (Prime, Schrödinger, LLC, New York, NY, USA, 2016) software, employing a crystal structure of L-ARG belonging to *Leishmania mexicana* (PDB ID 4IU1) due to the high sequence identity with *Leishmania amazonensis,* for which no crystal structure is available [18]. The output of this calculation is reported in Figure 1, Figure 2 and Figure 3.

The first analysis was performed on CAPA and its close derivative, compound **6** (Figure 1). They showed a comparable inhibitory potency against L-ARG (Table 1), and based on molecular docking studies they also shared a similar binding mode. In particular, both compounds are able to establish a series of polar and hydrophobic interactions within the binding site of L-ARG. The catechol moiety of both compounds is involved in a metal coordination bond by one of its hydroxyl groups (Figure 1A,B). Moreover, for CAPA and compound **6** the hydroxyl groups are able to form two H-bonds with the side chains of T257 and E288. Furthermore, the aromatic component of the catechol moiety established a double π-π stacking with H139 and H154. The amide portion, by its carbonyl group was be able to form an H-bond with the sidechain of S150, as observed by the docking output of both compounds. The terminal region of the selected compounds showed a different behavior. In particular, the phenyl ring of CAPA established only a π-π stacking with H139, while the phenol ring of compound **6**, in addition to the same π-π stacking described for CAPA, was able to form an additional H-bond with the backbone of P258 by its hydroxyl group. This additional contact could be the reason for the slight improvement of the potency of compound **6**. Accordingly, these patterns of interaction found for CAPA and compound **6** accounted for a docking score and an estimated ligand binding energy (CAPA, XP score = −7.68 kcal/mol; ΔG_bind_ = −70.58 kcal/mol; compound **6**, XP score = −7.32 kcal/mol; ΔG_bind_ = −75.39 kcal/mol) in line with our previous computational studies, performed on comparable compounds, and were in line with the micromolar inhibition of the enzyme.

Regarding the other two congeneric compounds (**13** and **14**) that act as competitive L-ARG inhibitors, the output of docking calculation is reported in Figure 2. In addition, in this case the catechol moiety is projected towards the reactive center of the enzyme. In fact, this region of both molecules is involved in a metal coordination bond by one of its hydroxyl groups. Moreover, this moiety of both compounds established the same contacts described for the above mentioned compounds (two H-bonds with the side chains of T257 and E288, and a double π-π stacking with H139 and H154). The amide portion of both compounds was able to form an additional contact with the amide portion belonging to CAPA and compound **6**. In fact, the carbonyl group can form an H-bond with the sidechain of S150, with the NH group H-bonding D194 (Figure 2A,B). Accordingly, these patterns of interaction found for compounds **13** and **14** accounted for a similar docking score and estimated ligand binding energy (compound **13**, XP score = −7.51 kcal/mol; ΔG_bind_ = −74.22 kcal/mol; compound **14**, XP score = −7.36 kcal/mol; ΔG_bind_ = −77.95 kcal/mol) highlighting that the introduction of the trimethoxybenzene moiety in compound **14**, replacing the phenyl ring of compound **13,** did not give any improvements in terms of activity. This was supported by the docking calculation, since the terminal portion of both compounds did not produce relevant contacts, with exclusion of the hydrophobic interaction with the sidechain of V149 (Figure 2).

The other competitive inhibitor **17** was also investigated by in silico studies (Figure 3). Interestingly, due to the reduction of the length and the increase of the flexibility of the molecule, compound **17** was only able to form the same contacts described for the catechol moiety belonging to the above reported compounds (metal interaction, two H-bonds with the side chains of T257 and E288, and a double π-π stacking with H139 and H154), while the other part of the molecule did not establish evident contacts. This led to a dramatic decrease of the inhibitory potency of the compounds against L-ARG (Table 1), also highlighted by the computational scores compared to the more potent inhibitor previously described (compound **17**, XP score = −6.63 kcal/mol; ΔG_bind_ = −69.58 kcal/mol).

For further rationalizing the experimental results we performed an extensive computational analysis in order to better understand the potential reasons for the selectivity of competitive inhibitors against L-ARG over B-ARG. To accomplish this task, we compared the molecular docking results described above with those obtained using the B-ARG homology model. The latter, due to the absence of a solved three-dimensional structure of this mammal arginase, were generated by means of homology modeling techniques. Accordingly, a homology model of B-ARG was built using the human arginase as template (PDB ID: 3VK2), employing Prime software (Prime, Schrödinger, LLC, New York, NY, USA, 2018), and as reported in the Material and Methods section. The resulting model was refined using the refinement protocol implemented in Prime, and evaluated by the Ramachandran plot obtained by RAMPAGE web server (Appendix A). The analysis revealed that the residues of protein were 93.9% in the favoured region of the plot, and 6.1% of the residues in the additional allowed region, with no residues in the disallowed region. So, all the residues of our refined B-ARG model sit in the allowed regions of the Ramachandran plot. This value is more than the cut-off value (96.1%) defined for the most reliable models [19]. Consequently, the quality of selected homology model was found to be satisfactory. Accordingly, the mentioned model was used in the subsequent computational analyses. We compared the binding sites of L-ARG and B-ARG by sequence alignment and SiteMap (SiteMap, Schrödinger, LLC, New York, NY, USA, 2018) calculations, the outputs are reported in Appendix A and Figure 4, respectively. The sequence alignment showed that although the general architecture and the organizational domains are maintained between the arginase enzymes, from *L. Mexicana* and *Bos Taurus*, the sequence identity and sequence similarity are not particularly high (38% and 52%, respectively). This reflects a relevant difference among the amino acids forming the enzymes, conferring different profiles to the binding site (Figure 4A). In particular, by superposing the structures we observed that a significant number of residues of the binding site of L-ARG are replaced in the binding site of B-ARG. The residues C156, A192, and D195 in L-ARG are replaced by Q143, D181, and P184, respectively, in the B-ARG. The residue A140 in L-ARG is replaced by T127 in the bovine enzyme. Notably, the residues V149 and S150 of L-ARG, composing an important interacting region, as found by docking calculation, are replaced by K136 and T137 in the B-ARG. This latter accounted for the relevant differences in the B-ARG binding site compared to the active site of L-ARG. In fact, as found by SiteMap calculation (SiteMap, Schrödinger, LLC, New York, NY, 2018; the output is reported in Figure 4B,C), the K136 caused a dramatic decrease of the total volume of the binding site of B-ARG, compared with the volume of the L-ARG (volume of B-ARG 94.31 A^3^, volume of L-ARG 158.84 A^3^). Accordingly, we hypothesized that these significant changes could affect the binding of some competitive inhibitors to B-ARG. To support our hypothesis, and for evaluating the influence of these changes in the binding of the selected compounds to B-ARG, we applied the same docking protocol used for L-ARG in order to compare the results of the docking calculation.

The mentioned homology model of B-ARG was employed for performing a computational analysis, based on molecular docking and ligand binding energy evaluation. The compound CAPA, and compounds **6**, **13**, **14,** and **17** were selected for the analysis in order to accomplish a direct comparison with the docking output obtained for L-ARG. The docking output is reported in Appendix A. Regarding CAPA and its close derivative compound **6** (Appendix A), which showed a comparable inhibitory potency against B-ARG to those found for L-ARG (Table 1), we also found a similar docking output. In fact, these compounds were found to bind B-ARG in a similar fashion as observed for L-ARG. Both compounds were accommodated into the B-ARG binding site with a similar shape as found for L-ARG (Figure 1), and with similar contacts. The catechol moiety of both compounds is involved in a metal coordination bond by one of its hydroxyl groups (Appendix A). Moreover, for CAPA and compound **6** the hydroxyl groups are able to form two H-bonds with the side chain of T246 and with the backbone of H141. Furthermore, the aromatic components of CAPA established two π-π stacking with H126 and H141, while compound **6** was only able to establish a π-π stacking with H141. The amide portion of CAPA was be able to form two H-bonds with the sidechain of K136 and D181 (Appendix A). The same portion of compound **6** only established a H-bond with the sidechain of K136 (Appendix A). As found for L-ARG, the terminal region of the selected compounds also showed a different behavior into B-ARG. In particular, the phenyl ring of CAPA established a π-π stacking with H126, while for the phenol moiety of compound **6** no stacking was observed. In addition, compound **6** was able to form an additional H-bond with the backbone of P247 by the hydroxyl group belonging to the phenol ring. This decrease in number of contacts found for compound **6,** compared to CAPA, could be the reason for its reduced potency against B-ARG. Accordingly, this pattern of interaction found for CAPA and compound **6** accounted for a docking score and an estimated ligand binding energy (CAPA, XP score = −7.61 kcal/mol; ΔG_bind_ = −68.51 kcal/mol; compound **6**, XP score = −6.54 kcal/mol; ΔG_bind_ = −60.73 kcal/mol) in agreement with the different inhibitory potency experimentally found for B-ARG.

The compounds **13** and **14,** highly active against L-ARG, showed a very weak inhibition against the bovine enzymes. We investigated by computational methodology the potential behavior of both compounds into B-ARG. The output is reported in Appendix A. In this case, contrary to the one found for CAPA and compounds **6,** we noted a totally different accommodation of **13** and **14** into B-ARG with respect to L-ARG. In fact, due to the volume decrease of the binding cavity and the presence of K136 at the entrance of the cleft, these two compounds were not able to establish relevant contacts for guaranteeing a strong inhibition of the bovine enzyme. In particular compound **13** (Appendix A) was able to form H-bonds with D128, with the hydroxyl groups belonging to the catechol moiety, but, due to the accommodation into the enzyme, this group seemed unable to form metal coordination bonds with the metal ions, since the distance from the two oxygens for the hydroxyl groups were found to be over 3 Å. The phenyl ring could establish a π-π stacking with H126, and NH from the central region could form a H-bond with the sidechain of D181. The other part of the molecule did not contribute to any interactions within the binding site, and it was found to be largely solvent exposed. The same was observed for compound **14** (Appendix A); in fact, no metal coordination bonds were detected, and only a H-bond with the backbone of H141, and a π-π stacking with H126 could be established by the catechol moiety. The amide portion could form a H-bond with the sidechain of K136, while the other part of the molecule was found to be largely solvent exposed, providing no contribution to the retrieved binding mode. Remarkably, all the poses found by docking calculation for both compounds were not able to produce any metal coordination bonds, and presented a similar shape, with the representative poses reported in Appendix A. Accordingly, this pattern of interaction found for compounds **13** and **14** accounted for a similar docking score and estimated ligand binding energy (compound **13**, XP score = −4.29 kcal/mol; ΔG_bind_ = −44.08 kcal/mol; compound **14**, XP score = −5.11 kcal/mol; ΔG_bind_ = −46.83 kcal/mol), highlighting the reduced affinity of both compounds for B-ARG.

Regarding compound **17,** due to the high flexibility of the molecule and the arrangement of the binding site, it showed an accommodation into the B-ARG binding site similar to that described for compounds **13** and **14**. Additionally in this case, the established interactions were H-bonds with D128 and D181, and a π-π stacking with H126, presenting a large region of the molecule exposed to the solvent (Appendix A). Accordingly, this pattern of interaction found for compound **17** accounted for a docking score and an estimated ligand binding energy (XP score = −5.44 kcal/mol; ΔG_bind_ = −40.15 kcal/mol) in agreement with the poor potency against B-ARG.

In addition, we investigated, through an extensive MD simulation, the behavior of the most selective compound of the series (compound **14**) into L-ARG and B-ARG, in order to better understand the reasons for its relevant selectivity (81-fold) towards L-ARG over B-ARG. The docking complexes, compound **14** into L-ARG (Figure 2A) and into B-ARG (Appendix A), were placed into a box filled with water, simulated by TIP3P model, and submitted to MD simulation, as reported in the Materials and Methods section. Notably, we performed five independent experiments for each complex, examining a trajectory of 200 ns with an aggregate time of 1 µs for each complex. This kind of extensive simulation was performed to better study the systems and to provide a reliable method for understanding in depth their behavior, excluding results obtained by chance. In particular, we observed that the complex compound **14/**L-ARG was found to be more stable than the complex compound **14/**B-ARG. In addition, the stability of the complex compound **14/**L-ARG was reached earlier than the complex compound **14/**B-ARG (25 ns vs. 150 ns respectively) (Figure 5A,D respectively). In addition, the root-mean-square fluctuation (RMSF) analysis Figure 5B–F demonstrated that the complex compound **14**/B-ARG (Figure 5E) presented highly fluctuant residues located in the binding site, as well as the ligand (Figure 5F), denoting a lower stability of the complex when compared to the RMSF analysis performed on the complex compound **14/**L-ARG (Figure 5B,C). This is absolutely in agreement with the weak inhibitory potency of **14** against B-ARG.

The investigation of the trajectory of compound **14/**L-ARG revealed that the main contacts found by docking were maintained as reported in Figure 6A. In particular, the metal coordination bond was maintained during all the simulations, as well as the main contacts retrieved by the docking calculation. In addition, compound **14** could form an additional H-bond with the sidechain of N152, and sporadically can target D137 and D243. The analysis of the trajectory of the complex compound **14**/B-ARG (Figure 6B) confirmed the reduced affinity of **14** for B-ARG, with a global number of contacts smaller than those found for compound **14** into L-ARG. The main contacts retrieved by the docking calculation were also found during MD simulation, with the exclusion of the H-bond with the sidechain of K136 that during the trajectory became sporadic. Other occasional contacts were found with D232 and D234, while the metal coordination bonds were not evident. To investigate the possibility of the formation of metal coordination bonds for compound **14** into both enzymes, we also monitored the distances from the hydroxyl groups to the metal ions for each complex, as described in the outputs in Appendix A for compound **14** into L-ARG, and in Appendix A for compound **14** into B-ARG. Accordingly, the main metal coordination bond found for compound **14** was maintained, and the distance was much lower than 3 Å during all of the simulation. In addition, we also observed that the other oxygen from the other hydroxyl group in some frames of the MD was able to coordinate the metal, since the distance found was under 3 Å (Appendix A). Regarding the compound **14** into B-ARG, the measurement confirmed the difficulty in reaching the reactive center of B-ARG, since the measurements provided were largely over 3 Å, with the exclusion of a short time in which one oxygen, from the hydroxyl group that is not involved in any H-bond in the docking calculation, was found under 3 Å from a metal ion (Appendix A). This analysis helped us to rationalize the experimental data, providing novel hints for the design of more potent and selective L-ARG inhibitors.

## 3. Discussion

We have highlighted *Leishmania amazonensis* as a potential target for the design of innovative antileishmanial drugs. First, we searched for L-ARG inhibitors in natural products, and described L-ARG inhibition by quercetin and others polyphenols [20,21,22]. At the same time, human arginase (h-ARG) was highlighted as a target in the vascular diseases [23].

The differences between L-ARG and h-ARG were previously described, leading to a possibility of selectively targeting parasite arginase [24]. The inhibitors developed to target mammalian arginase (ABH, BEC) have also been described as *Leishmania mexicana* arginase inhibitors [25,26]. The first synthetic L-ARG inhibitor, containing a thiosemicarbazide as a key pharmacophoric element, showed a selective parasite inhibition, but failed to kill the parasite efficiently in vitro [27,28]. Recently, a series of synthetic compounds, containing a phenilhydrazides moiety were described as a new pharmacophoric group that targets L-ARG, and kills the *L. amazonensis* promastigotes in vitro [29].

In this study, we tested a series of compounds primarily designed to target h-ARG, showing a unique compound named CAPA with activity against h-ARG [16]. Thus, the set of cinnamids containing a catechol group was tested, and revealed a selective potential against L-ARG. The compound CAPA showed a L-ARG inhibition in the low micromolar range (IC_50_ = 6.9 μM), and it was found to be nine-times more potent in inhibiting L-ARG, with respect to the h-ARG (IC_50_ = 60 μM) [16]. Compound **15** was the most active L-ARG inhibitor (IC_50_ = 1.3 μM). The differences between CAPA and compound **15** (Table 1) consist in the localization of the catechol group; CAPA is a cinnamid derived of caffeic acid (dihydroxycinnamic acid) and 2-phenylethylamine, while compound **15** is synthesized starting from cinnamic acid and 3,4-dihydroxyphenethylamine.

Although the compounds presented a similar structure, the cinnamide derivatives presented a different enzyme inhibition mechanism. CAPA also showed competitive inhibition of bovine arginase, and shared the same IC_50_ (6.9 μM) when tested against L-ARG. Both L-ARG and B-ARG were inhibited with comparable potency by the natural compound, chlorogenic acid (IC_50_ ~ 10 μM). Despite this similarity, the other compounds that inhibit L-ARG showed selective L-ARG inhibition, with a selectivity index varying from 4 (compound **6**) to 81 (compound **14**). These results reinforce the idea that selective L-ARG inhibition is possible.

All compounds showed arginase inhibition, and were then used in a trial against *L. amazonensis* promastigotes. The compound CAPA is the only one that showed activity against the parasite (IC_50_ = 80 μM). In addition, by an extensive computational protocol we investigated the binding modes of competitive inhibitors into L-ARG and B-ARG, for which we have built a homology model, in order to perform an in depth analysis of the binding sites, for highlighting the main differences. Moreover, the two enzymes were used, as mentioned above, in a rigorous computational analysis based on docking studies, and the evaluation of ligand binding energies, to provide the different behavior of the competitive inhibitors into both enzymes. We found that the volume of the binding site of the B-ARG was lower than the volume calculated for L-ARG, and this could probably affect the binding mode, and consequently the potency of the competitive inhibitors. Accordingly, with the exclusion of CAPA, the other compounds showed very different binding modes when they were docked into B-ARG, highlighting a general difficulty in reaching the reactive center and interacting with the metal ions, and reflecting a general decrease of the experimental inhibitory potency, as also indicated by the relevant decrease of the calculated ligand binding energies. Moreover, for providing an exhaustive analysis, with regards to the selectivity of some compounds against L-ARG, we selected compound **14**, that showed the highest selectivity index toward L-ARG, and performed MD simulations on two complexes obtained by the docking study, compound **14/**L-ARG, and compound **14/**B-ARG, to provide the basis for a different selectivity of compound **14** toward L-ARG over B-ARG. In summary, the computational efforts reported in this work, in addition to helping us to rationalize the experimental data, can provide novel hints for the design of more potent and selective L-ARG inhibitors.

## 4. Materials and Methods

### 4.1. Materials

The buffer CHES (2-(cyclohexylamino) ethane-sulfonic acid), L-arginine, CelLytic™ B, MOPS (4-morpholinepropanesulfonic acid), PMSF (phenyl-methyl-sulfonyl-fluoride) were purchased from Sigma-Aldrich. Reagents for urea analysis were obtained from Quibasa (Belo Horizonte, MG, Brazil). Cinnamide derivatives were synthesized as previously described [16].

### 4.2. Expression and Purification of Arginase

Recombinant L-ARG was expressed and purified as previously described [21]. Briefly, *Escherichia coli,* containing arginase gene cloned in plasmid DNA, was grown in SOB (Super Optimal Broth) medium at 37 °C until it reached absorbance 0.6 at 600 nm. Then the recombinant L-ARG expression was induced with a final concentration of IPTG (isopropyl β-D-1-thiogalactopyranoside) 1 mM and MnSO_4_ 10 mM for 3 h. The cells were harvested and lysed in 1 mL of buffer A (MOPS 100 mM, pH 7,2, NaCl 500 mM) containing PMSF (phenylmethanesulfonyl fluoride) 1 mM and 10% of CelLytic™ B. After lysis, the homogenate was centrifuged at 12,000× *g* for 10 min. Arginase in the supernatant was purified as previously described [30].

### 4.3. Arginase Inhibition and Kinetics

A stock solution (70 mM) was prepared in DMSO and stored at −20 °C for each compound just before the experiment. The inhibitory test was performed with 50 mM of L-arginine (pH 9.5), 50 mM CHES buffer (pH 9.5), and enzyme at 5 nM. The sample was incubated in a water bath at 37 °C for 15 min. Urea was quantified according to the Berthelot method [31], using commercial reagents. Briefly, the catalytic activity of the arginase reaction was stopped by transferring 2 μL of the reaction mixture into 150 μL of reagent I (20 mM phosphate buffer, pH 7, containing 60 mM Salicylate, 1 mM sodium nitroprusside, and >500 IU of urease). This mixture was incubated at 37 °C for 5 min. Next, 150 μL of reagent II (sodium hypochlorite 10 mM and NaOH 150 mM) was added, and the samples were incubated at 37 °C for 5 min. Measurement was taken at 600 nm using an Epoch 2 microplate reader (BioTek Instruments, Inc.). Two independent experiments were performed in triplicate until a coefficient of non-linear regression R^2^ ≥ 0.95 was obtained. The sigmoidal model (log IC_50_) was used to determine IC_50_ and maximum effect (E_max_) in GraphPad-Prism 6.01 software for Windows (San Diego, CA, USA).

### 4.4. Determination of the Constants Ki, Ki’ and the Mechanism of Inhibition

All reactions were performed in 50 mM CHES buffer, pH 9.5, containing variable concentrations of the substrate L-arginine (25, 50, and 100 mM) at pH 9.5. Inhibitors were tested at three different concentrations close to the IC_50_. All reactions were performed in triplicate in two independent experiments. Urea was quantified according to the Berthelot method [31] using commercial reagents as previously described. The constant Ki was determined for competitive and mixed inhibitors, and Ki’ was determined uncompetitive and mixed using a Dixon and Cornish–Bowden plots [32]. For non-competitive inhibitors, Ki and K’_i_ values were the same, and y = 0 was determined to obtain that value [20].

### 4.5. Promastigotes Culture

Promastigote forms of the strain MHO were inoculated initially in the concentration of 5 × 10^5^ parasites/mL in M199 medium (Life Technologie, Carlsbad, CA, USA), pH 7.4, supplemented with 10% of fetal bovine serum, (50 µg/mL) streptomycin, and (100 U) penicillin at 24 °C. Cells were grown until they reached the stationary phase. After this, 998 μL of the culture at 5.0 × 10^5^ cells/mL was incubated with 2 μL of the inhibitors (50 to 0.0005 mM) dissolved in DMSO to obtain final concentrations of 100 to 0.001 μM. Amphotericin B was used as a control of the cell growth inhibition, in a concentration varying from 60 to 0.0006 μM dissolved in DMSO. The test of inhibition of the parasite growth was performed during 78 h in the presence of the L-ARG inhibitors. After this inhibition period, the surviving cells were washed twice with HBSS (Gibco), centrifuged at 2000× *g* for 10 min, and supernatant was removed. To verify the parasite viability, over the pellet of cells about 1000 µL of M199 medium content was added, with 5 mg/mL of MTT (3-(4,5-dimethylthiazol-2-yl)-2, 5-diphenyltetrazolium bromide, a tetrazole) (Sigma-Aldrich, St. Louis, MO, USA). Surviving parasites were exposed to MTT during 12 h at 34 °C in a CO_2_ chamber. After this period, the cells were centrifuged for 10 min at 2000× *g* and the supernatant was removed. Formazan crystals that were formed in the cells were dissolved with DMSO, the absorbance was read using an Epoch 2 Microplate Spectrophotometer at 600 nm.

### 4.6. Computational Details

*Molecule preparation.* The three-dimensional structures of the L-ARG competitive inhibitor, CAPA, and compounds **6**, **13**, **14,** and **17** were built in Maestro molecular modelling environment (Maestro release 2016), using MacroModel software for minimizing the molecules (MacroModel, Schrödinger, LLC, New York, NY, USA, 2016), as reported by us [33,34], with OPLS-AA 2005 as force field, and a GB/SA model for simulating the solvent effects [35,36,37]. The PRCG method, with 1000 maximum iterations, and 0.001 gradient convergence threshold was used. In addition, LigPrep (LigPrep, Schrödinger, LLC, New York, NY, USA, 2016) was employed to refine the chemical structures [38].

*Protein preparation.* According to our previous works [15,17,18] we used the crystallized structure of the *L. mexicana* arginase (PDB ID: 4IU1) [25], available in PDB, for the computational studies. This structure was imported into Maestro suite and submitted to the protein preparation wizard protocol implemented in Maestro suite 2016 (Protein Preparation Wizard workflow 2016) for obtaining a reasonable starting structure for further in silico experiments [34,39].

*Homology modeling of bovine arginase.* The homology model of B-ARG was generated by following a computational protocol described by us [40,41]. The sequence of B-ARG was taken in FASTA format from UniProtKB (entry Q2KJ64). The homology model of B-ARG was built using Prime software (Prime, Schrödinger, LLC, New York, NY, USA, 2018). Suitable templates were searched by the BLAST homology search implemented in Prime. Human arginase (PDB ID 3KV2) [42] was selected for building the homology model. Due to the high sequence identity (90%) and sequence similarity (94%) we decided to adopt the single-template based alignment technique for building the three-dimensional model of B-ARG. During the calculation we included the metal ions from the template in the generated model. The best predicted model was further refined by Prime, with the implemented refinement protocol including loop refinement and whole structure minimization. The quality of the model was assessed by RAMPAGE webserver (http://mordred.bioc.cam.ac.uk/~rapper/rampage.php) (Appendix A). The model was submitted to Protein Preparation Wizard workflow 2016, as above described for L-ARG.

*Binding sites analysis.* The analysis of the binding sites of L-ARG and modelled B-ARG was performed by means of SiteMap (SiteMap, Schrödinger, LLC, New York, NY, USA, 2018), with the default settings, and focusing the calculation on the binding sites enclosing the metal ions.

*Molecular docking.* Docking experiments were performed by Glide (Glide, Schrödinger, LLC, New York, NY, USA, 2018), using the ligands and the proteins prepared as mentioned above with the Glide extra precision (XP) method. The energy grid was prepared using the default value of the protein atom scaling factor (1.0 Å), within a cubic box centered on the crystallized ligand nor-NOHA for L-ARG, while for the modelled protein the cubic box was centered on metal ions. As part of the grid generation procedure, metal constraints for the receptor grids were also applied [43,44]. After the grid generation, the ligands were docked into the enzymes, considering the metal constraints. The number of poses entered to post-docking minimization was set to 500, and the Glide XP score was evaluated.

*Estimated ligand binding energy.* The Prime/MM-GBSA method implemented in Prime software (Prime, Schrödinger, LLC, New York, NY, USA, 2016) computes the change between the free and the complex state of both the ligand and the protein after energy minimization. This technique was used on the docking complexes (L-ARG/inhibitors; B-ARG/inhibitors) presented in this study, and obtained by the molecular docking calculations. The software was employed to calculate the ligand binding energy (ΔG_bind_), as previously reported [15,17,18].

*Molecular dynamics simulation.* Molecular dynamics (MD) simulations studies were performed by means of Desmond 5.6 academic version, provided by D. E. Shaw Research (“DESRES”), using Maestro as a graphical interface (Desmond Molecular Dynamics System, version 5.6, D. E. Shaw Research, New York, NY, USA, 2018). The calculation was performed using the Compute Unified Device Architecture (CUDA) API [45] employing two NVIDIA GPUs [46]. The calculation was performed on a system comprising 56 Intel Xeon E5-2660 v4@2.00 GHz processors and two NVIDIA GeForce RTX 2070 GPUs. The complexes, L-ARG/compound **14** and B-ARG/compound **14**, obtained from docking calculation, were prepared by Protein Preparation Wizard protocol. The complexes were positioned into an orthorhombic box filled with water (TIP3P model). OPLS_2005 force field was used in MD calculation. The physiological concentration of monovalent ions (0.15 M) was simulated by adding Na^+^ and Cl^−^ ions. Constant temperature (300 K) and pressure (1.01325 bar) were employed with NPT (constant number of particles, pressure, and temperature) as ensemble class. RESPA integrator [47] was used in order to integrate the equations of motion, with an inner time step of 2.0 fs for bonded interactions and non-bonded interactions, within the short-range cut-off. Nose–Hoover thermostats [48] were employed for maintaining the constant simulation temperature, and the Martyna–Tobias–Klein method [49] was employed for controlling the pressure. Long-range electrostatic interactions were evaluated using the particle-mesh Ewald method (PME). The cut-off of 9.0 Å was used for van der Waals, and short-range electrostatic, interactions. The equilibration of the systems was performed with the default protocol provided in Desmond, which consists of a series of restrained minimizations and MD simulations used to slowly relax the system. By following this protocol, a single trajectory of 200 ns was obtained. We performed five independent MD runs for each mentioned complex, with an aggregate simulation time of 1 μs for improving the robustness of calculation, and for proposing a more reliable discussion. The trajectory files were investigated by simulation interaction diagram tools, simulation quality analysis, and simulation event. The described applications were used to generate all plots regarding MD simulations analysis included in the manuscript, as reported in the Results and Discussion section.

### 4.7. Data Analysis

Statistical analysis was performed by ANOVA and post hoc Tukey’s tests, using GraphPad-Prism 6.01 software for Windows (San Diego, CA, USA). For all tests *p* < 0.05 were considered significant.

## 5. Conclusions

The cinnamide derivatives tested in the present study were potent inhibitors of L-ARG, with a significant selectivity for this enzyme over the mammalian counterpart. Computational analysis provided the reasons for the selective inhibition of compound **14** (81 times selective for L-ARG over B-ARG) that could help scientists to rationally design selective compounds for L-ARG over mammalian isoform, paving the way for the development of innovative and selective drugs against L-ARG based on a cinnamide scaffold.

## Figures and Tables

**Figure 1 molecules-25-05271-f001:**
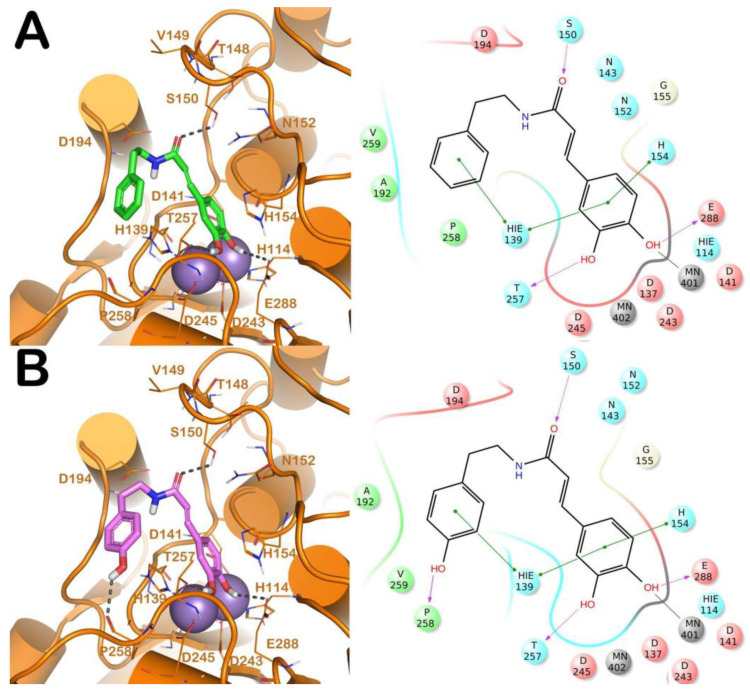
(**A**) Putative binding mode of CAPA (green sticks) into L-ARG binding site (PDB ID 4IU1; orange cartoon) as found by Glide software. Metals are represented as gray spheres. Key residues of the binding site are represented by lines. H-bonds are represented as black dotted lines, while the metal coordination bonds are represented by colored dotted lines. The ligand interaction diagram is reported on the right panel. (**B**) Putative binding mode of compound **6** (pink sticks) into L-ARG binding site (PDB ID 4IU1; orange cartoon) as found by Glide software. Metals are represented as gray spheres. Key residues of the binding site are represented by lines. H-bonds are represented as black dotted lines, while the metal coordination bonds are represented by colored dotted lines. The picture was generated by PyMOL (The PyMOL Molecular Graphics System, v1.8; Schrodinger, LLC, New York, NY, USA, 2015), while the ligand interaction diagram was generated by Maestro (Maestro, Schrödinger, LLC, New York, NY, USA, 2016).

**Figure 2 molecules-25-05271-f002:**
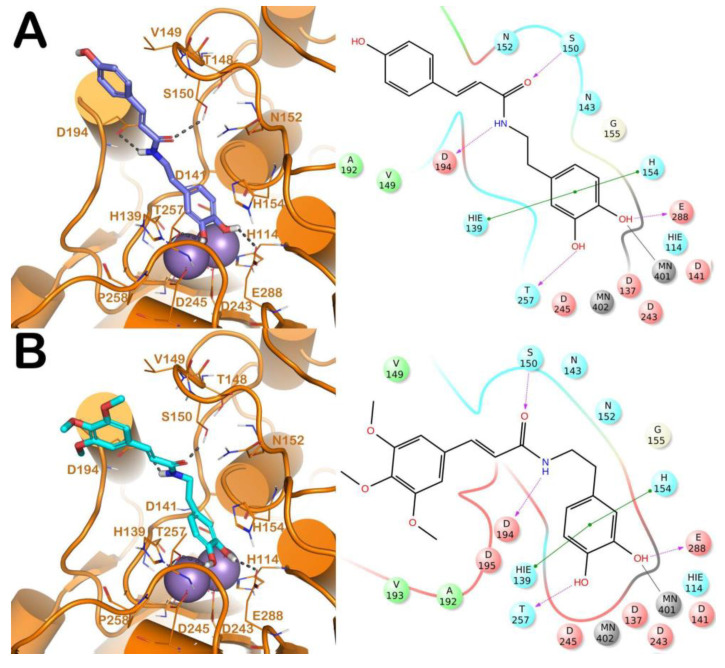
(**A**) Putative binding mode of compound **13** (blue sticks) into L-ARG binding site (PDB ID 4IU1; orange cartoon) as found by Glide software. Metals are represented as gray spheres. Key residues of the binding site are represented by lines. H-bonds are represented as black dotted lines, while the metal coordination bonds are represented by colored dotted lines. The ligand interaction diagram is reported on the right panel. (**B**) Putative binding mode of compound **14** (cyan sticks) into L-ARG binding site (PDB ID 4IU1; orange cartoon) as found by Glide software. Metals are represented as gray spheres. Key residues of the binding site are represented by lines. H-bonds are represented as black dotted lines, while the metal coordination bonds are represented by colored dotted lines. The picture was generated by PyMOL (The PyMOL Molecular Graphics System, v1.8; Schrodinger, LLC, New York, NY, USA, 2015), while the ligand interaction diagram was generated by Maestro (Maestro, Schrödinger, LLC, New York, NY, USA, 2016).

**Figure 3 molecules-25-05271-f003:**
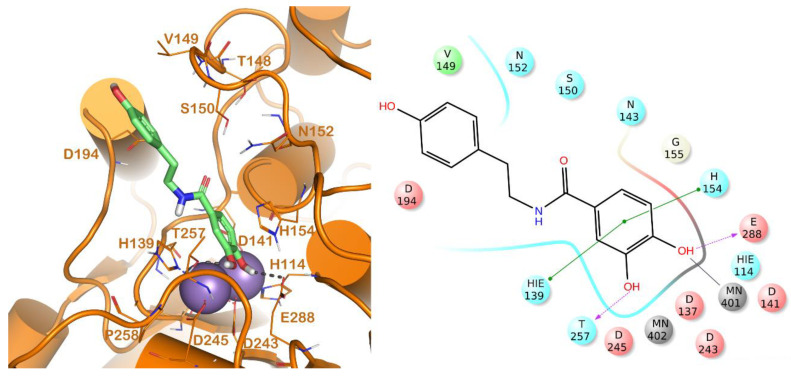
Putative binding mode of compound **17** (light green sticks) into L-ARG binding site (PDB ID 4IU1; orange cartoon) as found by Glide software. Metals are represented as gray spheres. Key residues of the binding site are represented by lines. H-bonds are represented as black dotted lines, while the metal coordination bonds are represented by colored dotted lines. The ligand interaction diagram is reported on the right panel. The picture was generated by PyMOL (The PyMOL Molecular Graphics System, v1.8; Schrodinger, LLC, New York, NY, USA, 2015), while the ligand interaction diagram was generated by Maestro (Maestro, Schrödinger, LLC, New York, NY, USA, 2016).

**Figure 4 molecules-25-05271-f004:**
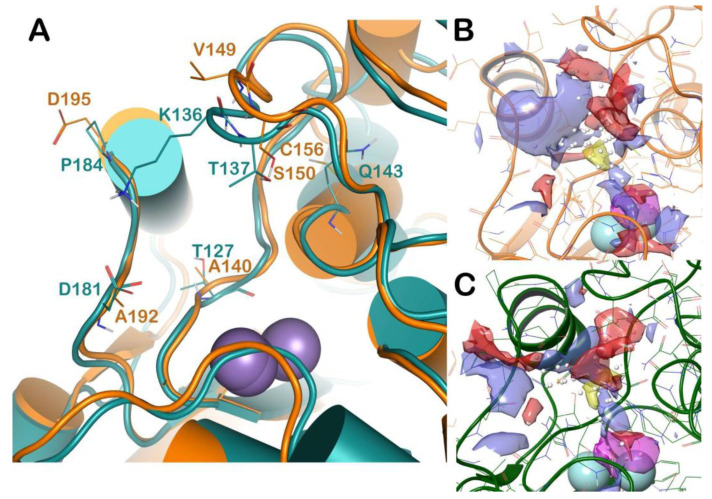
(**A**) Superposition of L-ARG (orange cartoon) with a homology model of B-ARG (dark green cartoon). The changed residues were highlighted, and reported as lines. The picture was generated by PyMOL (The PyMOL Molecular Graphics System, v1.8; Schrodinger, LLC, New York, NY, USA, 2015). (**B**,**C**) Output of the SiteMap calculation, highlighting the different surfaces of the two selected binding sites (L-ARG in orange cartoon; B-ARG in dark green cartoon). The pictures were generated by Maestro (Maestro, Schrödinger, LLC, New York, NY, USA, 2016).

**Figure 5 molecules-25-05271-f005:**
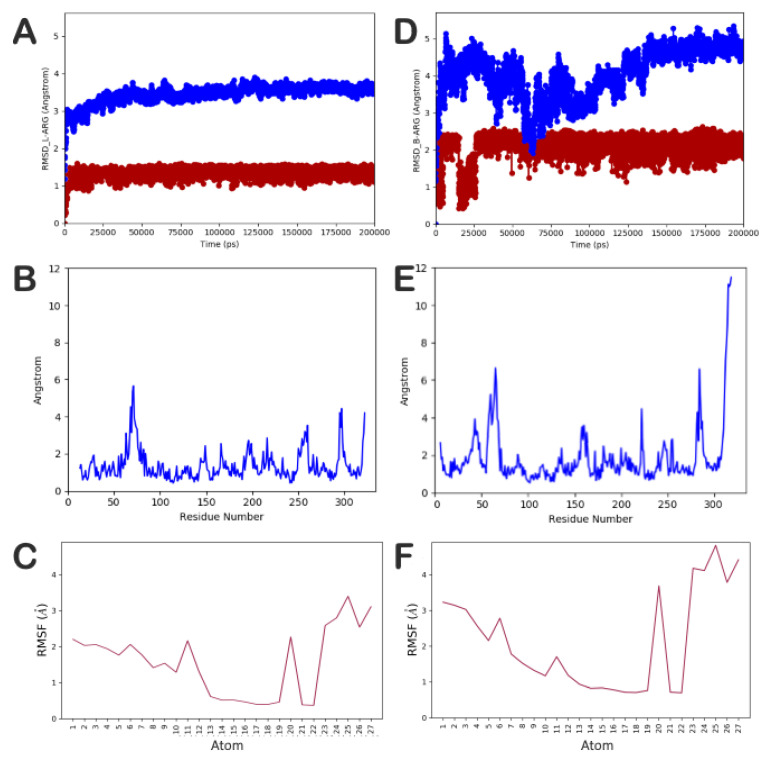
(**A**–**C**) Root-mean-square deviation (RMSD) and RMSF of the protein and the ligand, respectively, found for the complex compound **14/**L-ARG; (**D**–**F**) RMSD and RMSF of the protein and the ligand, respectively, found for the complex compound **14/**B-ARG. The pictures were generated by MD tools analysis implemented in Desmond.

**Figure 6 molecules-25-05271-f006:**
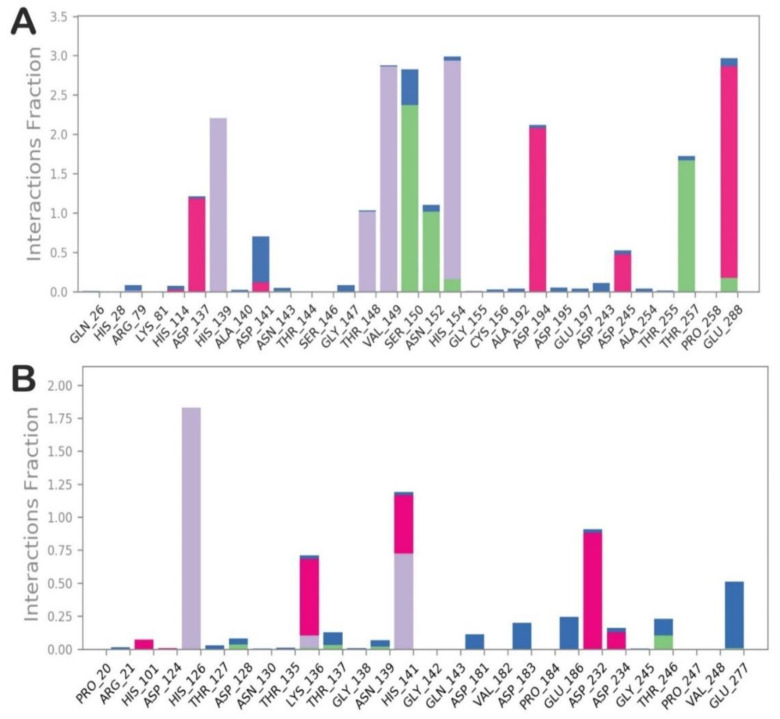
(**A**) Contacts established during the MD simulation by compound **14** within the binding site of L-ARG; (**B**) Contacts established during the MD simulation by compound **14** within the binding site of B-ARG. The pictures were generated by the MD tools analysis implemented in Desmond. Categorization of protein–ligand interactions into four types: H-bonds (green), hydrophobic (grey), ionic (magenta), and water bridges (blue).

**Table 1 molecules-25-05271-t001:** Arginase inhibition by cinnamide compounds. IC_50,_ E_max_, dissociation constant, and mechanism of inhibition.

Compound	Structure	IC_50_ (µM)	E_max_ (%)	K_i_ (µM)	K_is_ (µM)	Mode of Inhibition
**CAPA**	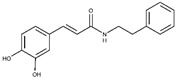	6.9 ± 0.7	79 ± 1	3.9 ±1.0	-	Competitive
**6**	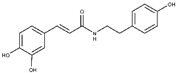	5.6 ± 0.4	82 ± 2	4.4 ± 1.1	-	Competitive
**11**	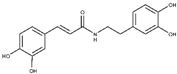	1.8 ± 0.1	81 ± 2	-	2.5 ± 0.2	Uncompetitive
**12**	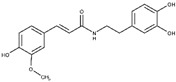	6.5 ± 1.4	75 ± 2	9.6 ± 0.7	9.6 ± 0.7	Noncompetitive
**13**	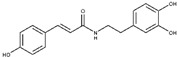	2.5 ± 0.2	82 ± 1	1.6 ± 0.2	-	Competitive
**14**	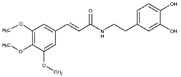	2.4 ± 0.9	81 ± 2	1.2 ± 0.3	-	Competitive
**15**	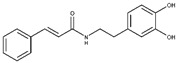	1.3 ± 01	80 ± 1	1.4 ± 0.2	5 ± 1	Mixed
**17**	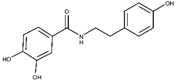	18 ± 3	68 ± 2	11.4 ± 0.7	-	Competitive
**18**	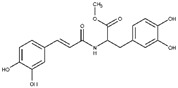	4 ± 1	89 ± 3	3.9 ± 0.4	27 ± 8	Mixed
**19**	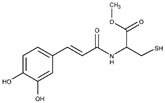	9 ± 1	90 ± 3	6.6 ± 0.3	28 ± 4	Mixed

IC_50_: half inhibitory activity; E_max_: maximal inhibitory activity at 100 μM.

**Table 2 molecules-25-05271-t002:** Comparison of IC_50_ inhibition of *Leishmania* and bovine arginase.

Compound	*L. amazonensis*	Bovine ^a^	Selective Index
**CAPA**	6.9 ± 0.7	6.9 ± 1.3	1
**6**	5.6 ± 0.4	22.1 ± 1.6	4
**11**	1.8 ± 0.1	114.9 ± 1.3	64
**12**	6.5 ± 1.4	198.7 ± 1.4	31
**13**	2.5 ± 0.2	170.4 ± 1.7	68
**14**	2.4 ± 0.9	193.6 ± 1.4	81
**15**	1.3 ± 0.1	39.3 ± 1.4	30
**17**	18 ± 3	175.3 ± 1.5	10
**18**	4 ± 1	41.9 ± 1.3	10
**19**	9 ± 1	37.0 ± 1.3	4
**Caffeic acid**	1.5 ± 0.3 ^b^	86.7 ± 8	58
**Chlorogenic acid**	8.3 ± 0.2 ^b^	10.6 ± 1.6	1.3

^a^ Reference [16]; ^b^ Reference [15].

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
