# Peer review of "Cinnamides Target Leishmania amazonensis Arginase Selectively"

_molecules, 2020, doi:10.3390/molecules25225271_

Round 1
Reviewer 1 Report
The present manuscript entitled “Cinnamides target arginase selectively” by da Silva et al. presents original results based on L-ARG inhibitors and merits publication in the Journal Molecules. However, some minor revision has to be addressed:
- In the abstract in vitro (line 25) must be in italics.
- Please, change the expressions best result (line 29, ) and the best (line 367) and use more appropriated scientific expressions. For example: most active …
- Authors should follow the same criteria with units. For example: minutes (line 412) and min (line 424). Choose one and use always the same.
- Authors should define abbreviations through the manuscript, mainly in the section Material and Methods. Examples: SOB (line 408), IPTG (line 410), PMSE (line 411).
- Authors mentioned that CAPA was the only compound that showed activity against the parasite (IC50= 80 µM) (lines 89-91 and lines 379-380) but, where are the activities of the other compounds against L. amazonensis in vitro?
Author Response
- In the abstract in vitro (line 25) must be in italics.
- Please, change the expressions best result (line 29, ) and the best (line 367) and use more appropriated scientific expressions. For example: most active …
- Authors should follow the same criteria with units. For example: minutes (line 412) and min (line 424). Choose one and use always the same.
- Authors should define abbreviations through the manuscript, mainly in the section Material and Methods. Examples: SOB (line 408), IPTG (line 410), PMSE (line 411).
The corrections 1-4 were made and were marked in red in the main text
- Authors mentioned that CAPA was the only compound that showed activity against the parasite (IC50= 80 µM) (lines 89-91 and lines 379-380) but, where are the activities of the other compounds against L. amazonensis in vitro?
The other compounds did not show any activity against the parasite. This information is highlighted on line 379:
"The compound CAPA is the only one that showed activity against the parasite (IC50 = 80 μM)"

Reviewer 2 Report
The researchers used the existing knowledge in the literature on arginase inhibitors, using classical biochemistry methods they have found that cinnamides derivate CAPA and other inhibit with high affinity and selectivity to arginase gene originating from Leishmania amazonensis. In order to understand the action's mode of the inhibitors and its selectivity to L-ARG, they used in silica tools to demonstrate that. The study is comprehensive, well written and welldone.
I Still have a question about the possible modeling for the other Arginase from other Leishmania species especially Leishmania donovani/infant complex
Author Response
The arginase Km from Leishmania amazonensis, Leishmania mexicana, and Leishmania infantum is the same. L. infantum and L. donovani have 95% identical amino acids while L. mexicana has 99% when compared with L. amazonensis. Thus the modeling is possible and could be used in drug development to target arginase in all species.